# Discoveries of Exoribonuclease-Resistant Structures of Insect-Specific Flaviviruses Isolated in Zambia

**DOI:** 10.3390/v12091017

**Published:** 2020-09-11

**Authors:** Christida E. Wastika, Hayato Harima, Michihito Sasaki, Bernard M. Hang’ombe, Yuki Eshita, Yongjin Qiu, William W. Hall, Michael T. Wolfinger, Hirofumi Sawa, Yasuko Orba

**Affiliations:** 1Division of Molecular Pathobiology, Research Center for Zoonosis Control, Hokkaido University, Sapporo 001-0020, Japan; christida@czc.hokudai.ac.jp (C.E.W.); m-sasaki@czc.hokudai.ac.jp (M.S.); h-sawa@czc.hokudai.ac.jp (H.S.); 2Hokudai Center for Zoonosis Control in Zambia, Research Center for Zoonosis Control, Hokkaido University, Lusaka 10101, Zambia; harima@czc.hokudai.ac.jp (H.H.); yeshita@czc.hokudai.ac.jp (Y.E.); yongjin_qiu@czc.hokudai.ac.jp (Y.Q.); 3Department of Para-Clinical Studies, School of Veterinary Medicine, University of Zambia, Lusaka 10101, Zambia; mudenda68@yahoo.com; 4Center for Research in Infectious Disease, University College Dublin, Dublin, Ireland; william.hall@ucd.ie; 5International Collaboration Unit, Research Center for Zoonosis Control, Hokkaido University, Sapporo 001-0020, Japan; 6Global Virus Network, Baltimore, MD 21201, USA; 7Department of Theoretical Chemistry, University of Vienna, 1090 Vienna, Austria; 8Research Group Bioinformatics and Computational Biology, Faculty of Computer Science, University of Vienna, 1090 Vienna, Austria

**Keywords:** Barkedji virus, Barkedji-like virus, dISFVs, 5′-UTR, 3′-UTR, RNA secondary structure

## Abstract

To monitor the arthropod-borne virus transmission in mosquitoes, we have attempted both to detect and isolate viruses from 3304 wild-caught female mosquitoes in the Livingstone (Southern Province) and Mongu (Western Province) regions in Zambia in 2017. A pan-flavivirus RT-PCR assay was performed to identify flavivirus genomes in total RNA extracted from mosquito lysates, followed by virus isolation and full genome sequence analysis using next-generation sequencing and rapid amplification of cDNA ends. We isolated a newly identified Barkedji virus (BJV Zambia) (10,899 nt) and a novel flavivirus, tentatively termed Barkedji-like virus (BJLV) (10,885 nt) from *Culex* spp. mosquitoes which shared 96% and 75% nucleotide identity with BJV which has been isolated in Israel, respectively. These viruses could replicate in C6/36 cells but not in mammalian and avian cell lines. In parallel, a comparative genomics screening was conducted to study evolutionary traits of the 5′- and 3′-untranslated regions (UTRs) of isolated viruses. Bioinformatic analyses of the secondary structures in the UTRs of both viruses revealed that the 5′-UTRs exhibit canonical stem-loop structures, while the 3′-UTRs contain structural homologs to exoribonuclease-resistant RNAs (xrRNAs), SL-III, dumbbell, and terminal stem-loop (3′SL) structures. The function of predicted xrRNA structures to stop RNA degradation by Xrn1 exoribonuclease was further proved by the in vitro Xrn1 resistance assay.

## 1. Introduction

Insect-specific viruses (ISVs) have been discovered worldwide and have been isolated from a range of mosquito species. These belong to a range of viruses in the order *Bunyavirales* and several families, including *Birnaviridae, Togaviridae, Rhabdoviridae, Reoviridae,* and *Flaviviridae.* ISVs were initially identified in the supernatants of *Aedes aegypti* cell lines and have been named the cell-fusing agent virus (CFAV) of the genus *Flavivirus* in the family *Flaviviridae* [1]. Viruses of the genus *Flavivirus* are mainly categorized based on their host range, which include mosquito-borne flaviviruses (MBFVs), tick-borne flaviviruses (TBFVs), no-known vector flaviviruses (NKVs), and insect-specific flaviviruses (ISFVs) [2].

Recently, identification, isolation, and phylogenetic analysis of ISFVs revealed two different groups: classic-insect specific flaviviruses (cISFVs) and dual-host affiliated insect-specific flaviviruses (dISFVs). The cISFVs were identified in and isolated from mosquitoes and are known to replicate only in mosquito-derived cell lines. While virus isolation has been reported in Australia, Brazil, Senegal, Japan, and several European countries [3], the role of cISFVs in MBFV infection and dissemination in mosquitoes is poorly understood. In particular, answers to the question of whether they can inhibit, support or are even unrelated to MBFV replication in mosquitoes remain elusive [1]. Similarly, several dISFVs have been discovered and isolated in mosquitoes. Although dISFVs only replicate well in mosquito-derived cell lines, they are phylogenetically more closely related to MBFVs than cISFVs [4]. Kenney et al. [5] hypothesized about three possibilities for the emergence of dISFVs: First, these viruses represent a distinct group of ISFVs, which did not evolve an ability to replicate in a vertebrate host; second, these viruses were part of a dual-host mosquito-vectored virus but have lost their ability to infect vertebrates; third, these viruses were part of the dual-host mosquito-vectored clade but the non-mosquito secondary host has not yet been identified.

In contrast to cISFVs, co-infection of dISFVs (Nhumirim virus (NHUV)) with MBFVs (West Nile (WNV), St. Louis Encephalitis (SLEV), Zika (ZIKV), and Dengue (DENV) viruses) in C6/36 cell lines have suggested that they could be suppressing MBFV production [5,6]. dISFVs have mainly been identified in *Aedes* spp. [7,8] and *Culex* spp. [9,10]. Both groups of these mosquito species are endemic in tropical and sub-tropical regions. The *Aedes* spp. mosquitoes are vectors for clinically important flaviviruses, such as DENV, Yellow fever virus (YFV) and ZIKV [11]. Similarly, *Culex* spp. mosquitoes are also distributed worldwide but are most prevalent in tropical and sub-tropical regions [12,13]. These mosquitoes are well known as vectors for encephalitic viruses, including WNV, SLEV, Rift Valley fever virus (RVFV), and Sindbis virus (SINV) [12].

Flaviviruses are characterized by an enveloped, single-stranded positive-sense RNA of some 10−11 kb genome length. The single-stranded open reading frame (ORF) encodes three structural proteins (capsid (C), pre-membrane/membrane (PrM/M), and envelope (E)) and seven non-structural proteins (NS1, NS2A, NS2B, NS3, NS4A, NS4B, and NS5) and is flanked by 5′- and 3′- untranslated regions (UTRs) [14]. The secondary structure and functions of the UTRs of flaviviruses have been well studied, specifically for pathogenic MBFVs and TBFVs [15]. Functional RNA elements are typically evolutionarily conserved at the level of their secondary structure, and manifested by structure-preserving nucleotide substitutions [16]. These covariation patterns are used by comparative genomics approaches for the identification of structurally homologous RNAs in phylogenetically related species [17].

The secondary structure of 5′-UTRs is conserved among many flaviviruses, and comprised of two stem-loop (SL) structures, SLA and SLB, which were shown to promote viral replication in DENV [18,19]. Downstream of the 5′-UTR, a conserved cis-acting capsid hairpin (cHP) structure is found at the beginning of the coding region and this is involved in virus cyclization and replication along with the 3′-UTRs [20]. In contrast, the flavivirus 3′-UTRs show a more varied pattern of evolutionarily conserved secondary structure elements, in particular in the SL and dumbbell (DB) elements, which are typically found as single or repeated entities [21]. Importantly, secondary structure elements in flavivirus 3′-UTRs, i.e., exoribonuclease-resistant RNA (xrRNA), can block RNA degradation by endogenous 5′-3′ exoribonucleases, such as Xrn1, resulting in the accumulation of sub-genomic flavivirus RNAs (sfRNAs) in the infected cells. While sfRNAs have been shown to interfere with the host immune responses, thereby modulating viral pathogenesis [22], studies on the 5′- and 3′-UTRs of dISFVs are limited. In particular, a functional characterization of predicted xrRNAs in dISFVs has not been previously reported to our knowledge.

Zambia, located in southern Africa, is divided into two ecological systems: dryland and wetland ecosystems. The wetland ecosystem is found in floodplains, lakes, swamps, and dambos, which are scattered across Zambia [23]. The Barotse floodplain is the biggest natural water source in Western province, and this provides an aquatic habitat for mosquito breeding [24].

To date in Zambia, human diseases related to flavivirus infection have not been reported. However, serological studies conducted on humans in Zambia have demonstrated antibodies against YFV, ZIKV, DENV, and WNV suggesting significant infections [25,26,27,28]. Recently, antibodies specific to ZIKV were also detected in non-human primates in Zambia [29]. Furthermore, our mosquito surveillance studies have shown that the first evidence of the circulation of WNV and the existence of a novel alphavirus in Zambia [30,31].

In this study, we report that our surveillance studies discovered two dISFVs, Barkedji virus (BJV) and a new virus tentatively named Barkedji-like virus (BJLV) from *Culex* mosquitoes. In parallel, to understand the evolutionary traits of dISFVs, we characterized the evolutionarily conserved, structural RNA elements in the untranslated regions of BJV and BJLV in silico by single sequence folding and consensus structure prediction. Furthermore, we focused on the functional characterization of BJV and BJLV xrRNA structures.

## 2. Materials and Methods

### 2.1. Study Site and Sample Collection

Mosquito samples were collected between 2012 and 2017 at several places in Zambia. Approval for the collection of Mosquitoes was obtained from the Zambia Wildlife Authority, now known as the Department of National Parks and Wildlife, Ministry of Tourism and Arts of the Republic Zambia following the study approval by the Excellence in Research Ethics and Science (ERES) converge ethics committee (IRB No: 00005948) and the National Health ethics research of the Ministry of Health. In total, six mosquito traps (three CDC light traps (John W. Hock Co, Gainesville, FL, USA) with CO_2_ produced by yeast fermentation and three BG-sentinel traps (Biogents AG, Regensburg, Germany)) were set up at several sites from afternoon until morning for five consecutive nights.

The collected mosquitoes were identified and classified by morphology using a stereomicroscope referred as the identification keys of African mosquitoes based on The Walter Reed Biosystematics Unit (http://www.wrbu.org/VecID_MQ.html), Kent R.J. [32], and Gillet J.D. [33]. In total, 1–40 individual mosquitoes were pooled based on the species and collection site. All pooled mosquitoes were stored at −80 °C. To confirm the mosquito species, PCR and sequencing of the cytochrome oxidase I (COI) gene were conducted [34].

### 2.2. Flavivirus Genome Screening

All mosquito pools were homogenized in minimum essential medium (MEM) containing 2% fetal bovine serum (FBS) using the BIOMASHER (Nippi, Tokyo, Japan). Total RNA was then extracted from 100 μL homogenates using the Direct-Zol kit (Zymo research, CA, USA) according to the manufacturer’s instructions. One-step RT-PCR was employed to detect the flavivirus genome using the One Step PrimeScript RT-PCR kit Ver.2 (Takara, Shiga, Japan) with pan-flavivirus primers sets (Appendix A), which targeted the NS5 gene of flaviviruses, as described by Patel et al. [35]. The cycling temperature was set as follows: 50 °C for 30 min, 94 °C for 2 min, 43 cycles of 94 °C for 30 s, 53 °C for 30 s and 72 °C for 30 s, and 72 °C for 5 min. For sequencing, the BigDye Terminator v3.0 Cycle sequencing kit was used on an ABI PRISM 3130 Genetic analyzer (Applied Biosystems, Foster City, CA, USA) with the Sanger sequencing analysis.

### 2.3. Virus Isolation

The remaining homogenates were filtered and inoculated into mosquito (C6/36) and mammalian (Vero and BHK-21) cells. The mosquito cells were maintained in MEM supplemented with 2% FBS, 2 mM L-glutamine, 0.1 mM non-essential amino acid, 100 U/mL penicillin, 100 μg/mL streptomycin, and 25 μg/mL gentamycin and incubated in 5% CO_2_ at 28 °C. The mammalian cells were maintained in MEM supplemented with 2% FBS, 2 mM L-glutamine, 100 U/mL penicillin, 100 μg/mL streptomycin, and 25 μg/mL gentamycin.

### 2.4. Next Generation Sequencing

RNAs extracted from the culture supernatant of BJV- or BJLV-infected C6/36 cells were utilized for the library preparation using the double-stranded cDNA synthesis kit (Takara) and the Nextera XT DNA Library Prep (Illumina, San Diego, CA, USA), followed by sequencing using a MiSeq Reagent Kit v3 (600 cycles) and Illumina MiSeq System (Illumina). CLC Genomics Workbench 10 (CLC Bio, Qiagen, Hilden, Germany) was used for de novo assembly data analysis.

### 2.5. Rapid Amplification cDNA Ends Analysis

The isolated viruses were propagated in C6/36 cells and incubated for 3 days. Viral RNA (vRNA) was extracted using the Direct-Zol kit (Zymo Research), followed by 5′-end identification by TaKaRa 5′-Full rapid amplification cDNA ends (RACE) Core Set (Takara) and 3′-end identification by SMARTer RACE 5′/3′ kit (Takara) according the manufacturer’s instructions.

Briefly, to identify 5′-end regions, the first strand cDNA was synthesized using the phosphorylated-gene specific primer BJVrace 5P-964R from the target vRNA through reverse transcription. The vRNA was degraded using RNase-H, and the first cDNA was circularized or concatemers were created using RNA ligase. Finally, DNA amplification was performed using PCR with virus-specific primer sets (Appendix A).

For 3′-end identification, the poly (A) tail was attached to the vRNA using *E. coli* poly (A) polymerase prior to cDNA synthesis. The cDNA was produced by incubating the poly(A)-tailed-vRNA with a 3′-CDS primer provided by the manufacture at 72 °C for 3 min and 42 °C for 2 min. Subsequently, after combining with a mixture containing DTT, dNTPs and RNase inhibitor, the mixture was incubated at 42 °C for 90 min and 70 °C for 10 min. Subsequently, Tris-EDTA buffer was added into the mixture.

The cDNA was then added into the RACE PCR master mix, which contained gene-specific primers, BJV40race 10,270 F or BJV31race 10,270 F (Appendix A). Thermal cycling was conducted as follows: 94 °C for 2 min, 30 cycles of 94 °C for 30 s, 68 °C for 30 s and 72 °C for 3 min, and 72 °C for 5 min. The PCR products (approximately 500 bp) were visualized on a 0.8% (*w*/*v*) agarose gel stained with ethidium bromide.

Finally, the PCR products for 5′- and 3′-RACE were sequenced using the Sanger sequencing method and aligned using ATSQ Ver 5.4.1 software. We identified the 5′- and 3′-UTR sequence using the Genetyx-MAC Ver.19 software.

### 2.6. Phylogenetic Tree Analysis

A phylogenetic tree was constructed using the Mega X software [36,37] by maximum likelihood method and Tamura-Nei model [38] and tested by bootstrap analysis with 1000 replications. Flavivirus genome data for constructing the tree was obtained from NIAID Virus Pathogen Database and Analysis Resource (ViPR) [39] through the web site at https://www.viprbrc.org/brc/vipr_genome_search.spg and Genbank databases (https://www.ncbi.nlm.nih.gov/genbank/). All genome sequences were aligned with MUSCLE [40].

### 2.7. Virus Susceptibility Test

We investigated whether the virus isolates could infect non-mosquito cell lines or only replicate specifically in mosquito cell lines. Therefore, we infected a confluent mammalian (Vero), avian (Duck embryo (ATCC CCL-141), CGBQ (ATCC CCL-169) and QT-6 (ATCC CRL-1708)), and insect (C6/36) cells in 12-well plates with 10 μL of 1 × 10^7^ copy μL^−1^ BJV or BJLV. These cells were purchased from ATCC (Manassas, VA, USA). Cell supernatants were collected at 6, 24, 48, and 72 h post infection.

Duck embryo cells were maintained in MEM supplemented with 10% FBS, L-glutamine, penicillin and streptomycin. CGBQ cells were maintained in Ham’s F-12K medium supplemented with 10% FBS, penicillin, and streptomycin; QT-6 cells were maintained in Ham’s F-12K medium supplemented with 10% FBS, 10% tryptose phosphate broth, penicillin, and streptomycin.

For the quantification of vRNA, the one-step qRT-PCR master mix was prepared using the TaKaRa One Step TB Green II kit (Takara) with BJV and BJLV-specific primer sets (Appendix A). The final concentration of the oligonucleotide primers employed for BJV and BJLV were as follows: 0.4 μM BJVzall 2622 F, 0.4 μM BJV L31 2622 F, and 0.4 mM BJVzall 2807 R. The qRT-PCR was run using the StepOnePlus instrument (Applied Biosystem).

### 2.8. Flavivirus Genome Data Set

For the comparative genomics screen, dISFV genomes were obtained from the public National Center for Biotechnology Information (NCBI) RefSeq (https://www.ncbi.nlm.nih.gov/refseq/) and Genbank (https://www.ncbi.nlm.nih.gov/genbank/) databases. We filtered for all complete virus genomes under taxonomy ID 11,051 (genus Flavivirus) as outlined previously [21]. For species without RefSeq annotation, we selected the longest complete genome from the Genbank set as representative sequence.

### 2.9. Characterization of Structurally Homologous RNAs

We employed an in silico comparative genomics assay, combining thermodynamic modeling by single sequence folding and consensus structure prediction as well as homology detection by utilizing covariance models (CMs). CMs are statistical models of RNA structure that extend classic Hidden-Markov-Models (HMMs) to represent sequence and secondary structure simultaneously [41]. They provide a powerful mechanism for the identification and characterization of homologous RNA structures and allow for a rapid screening of large RNA sequence databases to find even weakly conserved sequence-only or structurally homologous RNAs [42].

The Rfam database (https://rfam.xfam.org/) lists several functional flaviviral RNAs. In particular, CMs of the Rfam families RF00185 (Flavivirus 3′-UTR cis-acting replication element, Flavi_CRE), RF00465 (Japanese encephalitis virus hairpin structure), RF00525 (Flavivirus DB element), RF01415 (Flavivirus 3′-UTR stem loop IV) and RF00617 (Flavivirus capsid hairpin cHP) show hits in dISFV genomes, despite the fact that many of these models have been built from MBFV sequences. Following Ochsenreiter et al. [21], we adopted all hits produced by Rfam CMs for consistency reasons and used them to build dISFV-specific CMs.

In parallel, we computed structural multiple sequence alignments of dISFV 5′-UTR and 3′-UTR regions with *locarna* [43] and generated consensus structures with *RNAalifold* [44]. For the identification of conserved elements, locally stable secondary structures were computed using *RNALalifold* from the *ViennaRNA* package [45]. Novel CMs were then built for all conserved RNA elements in dISFVs.

We employed custom Perl scripts for the semi-automatic characterization and annotation of functional RNAs in dISFV UTR sequences, building on the ViennaRNA scripting language interface for thermodynamics calculations and the ViennaNGS [46] toolbox for extraction of genomic loci and processing of infernal output. All secondary structure plots were rendered using the RNAplot utility [45].

### 2.10. In Vitro Xrn1 Resistance Assay

The ability of BJV and BJLV xrRNA-like structures to block exoribonuclease 1 (Xrn1) was analyzed by an in vitro Xrn1 degradation assay as described by Dilweg et al. [47] and Chapman et al. [48]. Briefly, the template of BJV and BJLV xrRNA sequences (BJV 10,450–10,570 nt, BJLV 10,422–10,558 nt) were produced by PCR using PrimeSTAR GXL DNA polymerase (Takara) with specific primer sets with T7 promoter listed in Appendix A. Thermal cycling was conducted as 5 cycles of 98 °C for 10 s, 55 °C for 15 s and 68 °C for 30 s followed by 30 cycles of 98 °C for 10 s, 60 °C for 15 s and 68 °C for 30 s, and 68 °C for 5 min. After confirming the sequence by Sanger sequencing, in vitro transcription was conducted using MEGA script T7 (Ambion, CA, USA) following manufacturer’s instruction to obtain 3′-UTR xrRNA constructs.

The purified 3′-UTR RNA was then folded at 90 °C for 2 min and 20 °C for 5 min. Xrn1 digestion was performed with 2 µg RNA in 10× NEB3 buffer (100 mM NaCl, 50 mM Tris-HCl, 10 mM MgCl_2_, 1 mM DTT), 0.25 unit of 5′pyrophosphohydrolase (RppH) (NEB, Japan) and 1 µL of 31merRNA was added as control. The RNA mixture then divided into two tubes, 1 unit of Xrn1 (NEB, Japan) was added into one tube and the other tube was used as negative control with total volume of reaction was 20 µL. Thereafter, the tubes were incubated at 37 °C for 2 h. After incubation, the reaction was stopped by adding equal volume of 2× loading buffer containing formamide and incubate at 80 °C for 5 min. Finally, the samples were run onto 15% TBE-urea gel (Invitrogen, CA, USA) in 1× TBE buffer with pre-run stage at 180 V for 30 min and sample running at 180 V for 90 min. The gel was visualized under UV light after staining with ethidium bromide for 10 min.

### 2.11. Xrn1 Stop Point Identification

The RNA products of RNA degradation assay were extracted from the gel using ZR small-RNA PAGE recovery kit (Zymo research) following the manufacturer’s protocol. Then, reverse transcription was performed using SuperScript IV Reverse Transcriptase (Invitrogen) with approximately 2 pmol RNA. The RNA annealing was conducted using 5′-FAM-labeled primer (Appendix A) in the mixture of deoxyribonucleoside triphosphates (dNTPs), dideoxynucleoside triphosphates (ddNTPs) and RNAse free-water, then incubate at 95 °C for 5 min and 65 °C for 5 min. After that, reverse transcription mixture (5× buffer, DTT, SS IV, and RNase inhibitor) was added into the primer-RNA hybrid tubes and incubate at 50 °C for 10 min and 80 °C for 10 min. Finally, the reaction was stopped by adding the equal volume of the mixture of 95% formamide and 10 mM EDTA and incubate at 80 °C for 5 min.

The cDNA product was then analyzed using 15% urea-PAGE following previous protocol for pre-run stage and the running stage was performed at 180 V for 270 min. The gel was then visualized using ChemiDoc Touch imaging system (Bio-Rad Laboratories, CA, USA) and Image Lab Ver. 5.2 software (Bio-Rad) was used for image processing.

## 3. Results

### 3.1. Flavivirus Genome Detection and Virus Isolation

Mosquitoes were collected from various districts in Zambia from 2012 to 2017. In 2017, we collected in total 3304 female mosquitoes which were tested in 207 pools. Employment of a pan-flavivirus RT-PCR screening assay [35] led to the identification of partial NS5 gene sequences, which shared homology with BJV in one pool of *Culex quinquefasciatus* (zmq17L31) collected in Livingstone (Southern province), two pools of *Culex univittatus* (zmq17M54 and zmq17M115), and one pool of *Culex annulioris* (zmq17M40) collected in the Mongu districts (Western province).

The sequence homology of 250 bp RT-PCR products of the BJV NS5 gene was 99% for zmq17L31 and 81% for zmq17M40, zmq17M54, and zmq17M115 (Table 1). We further attempted to isolate the viruses using C6/36, Vero, and BHK-21 cells from the filtered homogenates of positive mosquito pools. Virus propagation could only be detected in C6/36 cells using RT-PCR and infection did not produce a clear cytopathic effect in C6/36 or the mammalian cells.

We employed next generation sequencing (NGS) to identify the whole-genomic sequences of the isolated viruses from mosquito lysates. Based on the NGS results, two different viruses were identified, with a nucleotide identity of 96% (zmq17L31) and 75% (zmq17M40, zmq17M54, and zmq17M115) compared to the BJV originally isolated in Israel (GenBank accession no. KC496020). Based on the flavivirus classification by Kuno et al. [49], isolate zmq17L31 was considered a strain of BJV (now named BJV Zambia) and isolates zmq17M40, zmq17M54, and zmq17M115 were novel flavivirus species which we have tentatively named Barkedji-like virus (BJLV). For further characterization of BJLV, we employed isolate zmq17M40.

### 3.2. Barkedji-Like Virus (BJLV)

The BJLV genome has 10,885 nucleotides (nt) length of RNA genomes, consisting of 10,320 nt ORF and flanked by 103 nt in the 5′-UTR and 462 nt in the 3′-UTR. BJLV shares 76% and 66% nucleotide identity with BJV Zambia and NHUV, respectively. The polyprotein of BJLV would be expected to have 3439 amino acids with three structural proteins (C, PrM/M, and E) and eight non-structural proteins (NS1, NS2A, NS2B, NS3, NS4A, 2K, NS4B, and NS5). Cleavage sites predictions of BJLV were determined by comparison with other BJV isolates: BJV Israel (GenBank accession no. KC496020.1), BJV Oman (GenBank accession no. AUS91146.1), and BJV Zambia (Table 2).

Pairwise amino acid comparison ranged from 62.6% (NS4A) to 77.4% (NS5) for NHUV (GenBank accession no. KJ210048) and from 81.5% (NS2A) to 92.4% (NS3) for BJV Zambia (Table 3). The complete genome sequence was submitted to GenBank with accession no. LC497469.

A phylogenetic tree analysis of the full-length nucleotide sequence revealed that BJLV is related to BJV, NHUV, and Nounané virus (NOUV). This virus group share ancestral roots with MBFVs, such as Dengue and the Yellow fever virus complex, but is distinct from cISFVs, such as CFAV (Figure 1).

### 3.3. Barkedji Virus (BJV)

The whole RNA genome of BJV Zambia (GenBank accession no. LC497470) is 10,899 nt long and is flanked by 102 nt in the 5′-UTR, 483 nt in the 3′-UTR, with 10,314 nt in the ORF. BJV Zambia shares 96.2% and 99.5% nucleotide and amino acid identity, respectively, with BJV Oman (GenBank accession no. AUS91146.1). This virus is almost identical to the BJV isolated in Israel, with a nucleotide identity coverage of up to 96.8%. The protein identity comparison between BJV Zambia and other flaviviruses are shown in Appendix A.

### 3.4. BJV Zambia and BJLV Growth in Mosquito and Vertebrate Cell Lines

To analyze the susceptibility of BJV Zambia and BJLV in mammalian and avian cell lines, the isolated viruses were inoculated in C6/36, duck embryo, CGBQ, QT-6 and Vero cells. Production of viral genome RNA for BJV Zambia and BJLV were detected only in the C6/36 cells. In the other cell lines, the number of viral RNA (vRNA) copies was gradually decreased (Figure 2).

### 3.5. Secondary Structure Analysis of 5′-UTRs

Single sequence predictions of the 5′-UTRs of both BJV Zambia and BJLV revealed a pattern of secondary structures similar to the MBFVs. In particular, homologs of the functional elements SLA and SLB, which have been attributed to promoting viral replication in Dengue viruses [18,19], were found at the 5′ end of both virus isolates. While the SLA element is located entirely within the 5′-UTR, the SLB element overlaps the canonical start codon (Figure 3). Interestingly, the SLB element in dISFVs appears longer than homologous structures in the MBFVs.

Immediately downstream of the SLB element, a cHP structure is present within the capsid (C) protein nucleotide sequence. Being located between two AUG codons, the cHP structure has been shown to mediate translation initiation by modulating start codon selection, while also being required for virus replication [50,51].

To support our findings by patterns of covariation, we followed the comparative genomics approach described in Ochsenreiter et al. [21] and de Bernardi Schneider et al. [21,52], i.e., we sought for structurally homologous RNAs in phylogenetically related viruses. Figure 4 shows the predicted consensus structures of the conserved elements SLA, SLB and cHP, which have been computed from structural multiple sequence alignments of homologous regions in related viruses. Here, the rich coloring of base paired columns indicates a high degree of covariation, thus supporting structural conservation.

### 3.6. Secondary Structure Analysis of 3′-UTRs

The phylogenetic proximity of the dISFV group to the well-studied group of MBFVs allowed us to assess the overall architectural traits of dISFV 3′-UTR in a recent study by means of Rfam CMs [21]. In the present study, the availability of additional sequence data allowed for a more fine-grained investigation of the individual 3′-UTR elements, which revealed a varied pattern of evolutionarily conserved, functional RNAs. Single sequence folding of the BJV/BJLV 3′-UTR regions, together with consensus structure predictions provided evidence for the presence of a cascade of conserved RNA structures, which have been associated with qualitative protection of downstream nucleotide sequences against exoribonuclease degradation. These RNAs, termed exoribonuclease-resistant RNAs (xrRNAs) have been described and verified in several different MBFVs [48,53,54,55,56,57] and, subsequently, in TBFVs, ISFVs and NKVs [58].

Figure 5 shows secondary structure predictions of the 3′-UTR regions of different flaviviruses. Different naming and numbering schemata were proposed for these stem-loop like structures in the literature, rendering comparison among different flavivirus species difficult. We therefore adopted a functionally descriptive naming system here by designating experimentally verified exoribonuclease-resistant RNAs as xrRNAs, and in silico-predicted structures that are homologous to experimentally verified exoribonuclease-resistant stem loop structures as xrRNA-like (xrRNA*) elements. MBFV xrRNAs are referred to as SL-II and SL-IV.

Figure 6 shows secondary structure predictions of the entire 3′-UTRs of BJV and BJLV. The terminal regions of the genomes can be subdivided into three autonomously folded parts, i.e., domains I-III, which typically contain one or two copies of conserved structured elements in many flaviviruses.

Domain I is located at the beginning of the 3′-UTR, downstream of the translation stop codon, and commonly appears as variable region. Here, variability manifests as an intrinsic unstructuredness at the beginning of the region, where BJV shows a longer unstructured region than BJLV. Downstream of the unstructured region, a three-way junction element is present, which is structurally homologous to known xrRNAs [21] and whose exoribonuclease stalling capacity is reported here. Figure 7a shows the consensus structure prediction of this element with xrRNA-like elements in other dISFVs. Notably, the closing stem of this element is longer in BJV Zambia and BJLV than in other dISFVs (Figure 5 and Figure 6).

Downstream of the xrRNA element, and yet within domain I, a hairpin of approximately 70 nt length, which is interspersed with multiple bulges and interior loops is found in both isolates. Comparison with other flavivirus 3′-UTRs revealed that this structure is essentially homologous to a long hairpin structure, which was initially shown in WNV [54] and termed SL-III. No biological function has been reported for this structure so far, however, the adjacent xrRNAs SL-II and SL-IV in WNV provide qualitative protection against exoribonuclease degradation [55]. Interestingly, among dISFVs the SL-III element is only conserved in BJV, BJLV, and NOUV. A consensus structure prediction of SL-III within dISFVs is shown in Figure 7b. The secondary structure comparison of 3′-UTRs is shown in Figure 5 and Figure 6.

Domain II of the BJV/BJLV 3′-UTRs contains small hairpin structures as well as a single copy of a canonical DB element, which is found in many flaviviruses [15,59]. Figure 7c shows the consensus structure prediction of the DB elements found in dISFVs, highlighting the previously observed pattern of significant covariations in the proximal stem-loop (positions 6–50), and almost perfect sequence conservation in the distal stem-loop (positions 50–70). A defining feature of domain III is the presence of the small hairpin (sHP) and a terminal 3′SL structure, which have been shown to be essential for virus replication [60]. Both show rich covariation patterns in the consensus structure prediction (Figure 7d).

### 3.7. xrRNA Structure of BJV and BJLV Can Block Xrn1 Enzyme

Our in silico analysis suggested that both BJV and BJLV have a three-way junction element that is structurally homologous to xrRNAs known in MBFVs. To investigate whether the xrRNA structures of BJV and BJLV have the same capability to stall Xrn1 we challenged the xrRNA structure (BJV 3′-UTR xrRNA and BJLV 3′-UTR xrRNA ∆HP), which were produced by in vitro transcription, with Xrn1 enzyme [Figure 8a(1,3)]. Both BJV and BJLV xrRNA structures blocked RNA degradation by the enzyme (Figure 8b).

In addition, we examined whether the hairpin structure located upstream of the BJLV xrRNA contributes to stopping Xrn1 (BJLV 3′-UTR xrRNA) (Figure 8a(2)). Our data show that the RNA product was similar to BJLV 3′-UTR xrRNA ∆HP. These results highlight that the hairpin structure was degraded by Xrn1 enzyme activity.

### 3.8. Determination of Xrn1 Enzyme Stop Site

The RNA product was then used for further analysis to determine the stop site of Xrn1 degradation. Reverse transcription was conducted to determine the RNA sequence, followed by nucleotide separation using urea-PAGE gel. Lastly, the nucleotide was detected by the emission of fluorescence signal. The results showed that the Xrn1 enzyme activity was stopped at A10,483 three nucleotides upstream of the BJV 3′-UTR xrRNA (Figure 8c,e) and was terminated at A10,471, one nucleotide ahead of the BJLV 3′-UTR xrRNA (Figure 8d,f).

## 4. Discussion

Based on the phylogenetic analysis, BJV Zambia and BJLV were grouped into the dISFV clade along with NHUV, NANV, NOUV, and Kampung Karu virus (KPKV), forming a monophyletic group related to the MBFVs (Figure 1). Notably, dISFVs represent a group of insect-specific flaviviruses, which are genetically distant from cISFVs [4]. BJV was initially identified in the Barkedji area of Senegal in 2007 (unpublished), and the virus genome was first characterized in Israel in 2011 [10]. BJV has also been identified in Oman (unpublished), Senegal, and the United Arab Emirates [61,62]. This virus has been mainly identified in *Culex* mosquito species, such as *Culex perexiguus* [10,62], *Culex neavei* [61], and *Culex quinquefasciatus*, with the exception of *Aedes sudanensis* in Senegal [61]. Previously, Garcia-Rejon et al. reported that 82% engorged female *Culex quinquefasciatus* infested on birds and the remaining 18% fed on mammals [63]. To examine the possibility that birds act as possible natural reservoirs of BJV Zambia and BJLV, which were isolated from *Culex* spp. similar to WNV, we tested the susceptibility of these viruses to avian cell lines, together with C6/36 and Vero cells. However, it could be shown that BJV Zambia and BJLV replicated exclusively in mosquito cells but not in bird and mammalian cells (Figure 2). Our findings are similar to those described in a previous report [10], which demonstrated that BJV Israel only replicated in mosquito cells. Also, other dISFVs, i.e., NHUV and NOUV, have been reported to replicate well in mosquito-derived cells but not in mammalian cells [9,64]. Further research on dISFVs are needed to investigate the evolution and origins of *Flavivirus*, and especially the emergence of MBFVs.

We characterized conserved RNA secondary structure elements in the untranslated regions of BJV Zambia and BJLV. Specifically, we employed RNA structural alignments to compute consensus structures and derived covariance models, which were used to complement the picture of evolutionary conservation in flavivirus UTRs.

A comparison of predicted 5′-UTR structures of BJV and BJLV with previously annotated elements in other dISFVs revealed a high degree of covariation, which is a strong evolutionary trait of the structural conservation of functional RNA elements. Our data provide evidence that BJV and BJLV conserve canonical 5′-UTR SLA and SLB elements, which can be found in many MBFVs and have been attributed to promoting viral replication in Dengue viruses [18,19]. The SLA consensus structure (Figure 4a) has a large central multi-loop that contains several gap symbols. This is caused by a variable length of this unpaired region in some related viruses, in particular, Ecuador Paraiso Escondido virus (EPEV), and partially MMV and DONV. The majority of sequences exhibit gap symbols in this part of the alignment, resulting in the given consensus structure. A conserved SLB element is found downstream of the SLA structure in all viruses analyzed in this study. The cHP hairpin element is found at the beginning of the capsid protein coding region in both BJV and BJLV.

BJV and BJLV phylogenetically form an independent group which is closely related to the MBFVs, with the exception of Yellow Fever virus complex. This proximity is well represented in the MBFV-like organization of conserved RNA elements in the 3′-UTR of these viruses (Figure 10). While the 3′-UTRs of BJV Zambia and BJLV, which are 483 nt and 462 nt, respectively, are the longest among all known dISFVs, they contain a set of conserved elements that have been associated either with pathogenicity or with specific roles in the flavivirus life cycle. Our comparative genomics screen revealed structural homologs to previously characterized, functional structures in the 3′-UTRs of BJV and BJLV, including the xrRNA, DB and terminal 3′SL elements [58]. The structural alignments and consensus structure predictions of these elements exhibit rich covariation support (Figure 7). Experimental verification of the predicted exoribonuclease-stalling activity revealed that the xrRNA elements of BJV and BJLV could halt degradation by Xrn 1. The presence of functional xrRNA in BJV and BJLV suggest that dISFVs have potency to produce sfRNA which could alter the cellular immune response and promote flavivirus transmission by mosquitoes [65]. Further investigation is needed to investigate the formation of the sfRNA and their molecular interactions with host factors in dISFVs.

The diversity of xrRNA structure in genus *Flavivirus* has been investigated by MacFadden et al. [58], who reported different types of xrRNA structures that could stop several types of exoribonucleases. It was shown that the production of sfRNA was not due to specific interactions of proteins and RNA but more likely mediated by the xrRNA structure itself, representing a general mechanical block for exoribonucleases. This led to a classification of xrRNAs in flaviviruses, with xrRNAs in cISFVs, MBFVs and several NKVs, i.e., Yokose virus, representing class 1 flavivirus xrRNA. The class 2 flavivirus xrRNA was included in the TBFVs and NKVs (Modoc virus, Montana myotis leukoencephalitis virus and Apoi virus) [58]. Here, we classify BJV and BJLV xrRNAs as class 1 flavivirus xrRNA, characterized by the Xrn1 stop site located upstream of the xrRNA structure (Figure 9).

In addition, we could predict the presence of a stem-loop structure (SL-III), which has not been previously functionally characterized in dISFV 3′-UTRs. Structural alignments of the relevant regions identified by CMs have led us to propose that an SL-III element is conserved in the 3′-UTRs of BJV, BJLV, and NOUV, but not in any other known dISFV species (Figure 7b and Figure 10). This is particularly interesting since SL-III hairpins have been predicted in some species of the Japanese encephalitis virus complex.

The present data do not allow us to form a conclusion on the evolutionary emergence of the SL-III element in distinct branches of the flavivirus phylogenetic tree. One possibility could be that co-infection with different flaviviruses, and subsequent recombination events resulted in increased fitness and the fixation of an SL-III element in ancestral viral species. Successive gene loss events by purifying selection could be one explanation for the presence of SL-III elements in divergent clades of the flavivirus tree today. In this regard, a functional characterization of SL-III and its association with specific roles in the viral life cycle will help make the underlying evolutionary traits more accessible.

## 5. Conclusions

In this study, we isolated and identified a new strain of Barkedji virus (BJV Zambia) and a novel flavivirus tentatively named Barkedji-like virus (BJLV). Both viruses were assigned to dISFVs based on a phylogenetic analysis. We demonstrated that BJV and BJLV could replicate effectively in mosquito cell line (C6/36) but not in mammalian and avian cell lines. The in silico investigation of the secondary structure of the 5′- and 3′-UTRs of BJV and BJLV showed that both viruses have canonical SL structure on the 5′- UTR, while the 3′-UTRs contain RNA elements with structurally homology to xrRNA, DB and terminal 3′-SLs found in MBFVs. Furthermore, both BJV and BJLV xrRNAs were classified as class 1 flavivirus xrRNA with the Xrn1 degradation stop point being located upstream of the xrRNA structures.

## Figures and Tables

**Figure 1 viruses-12-01017-f001:**
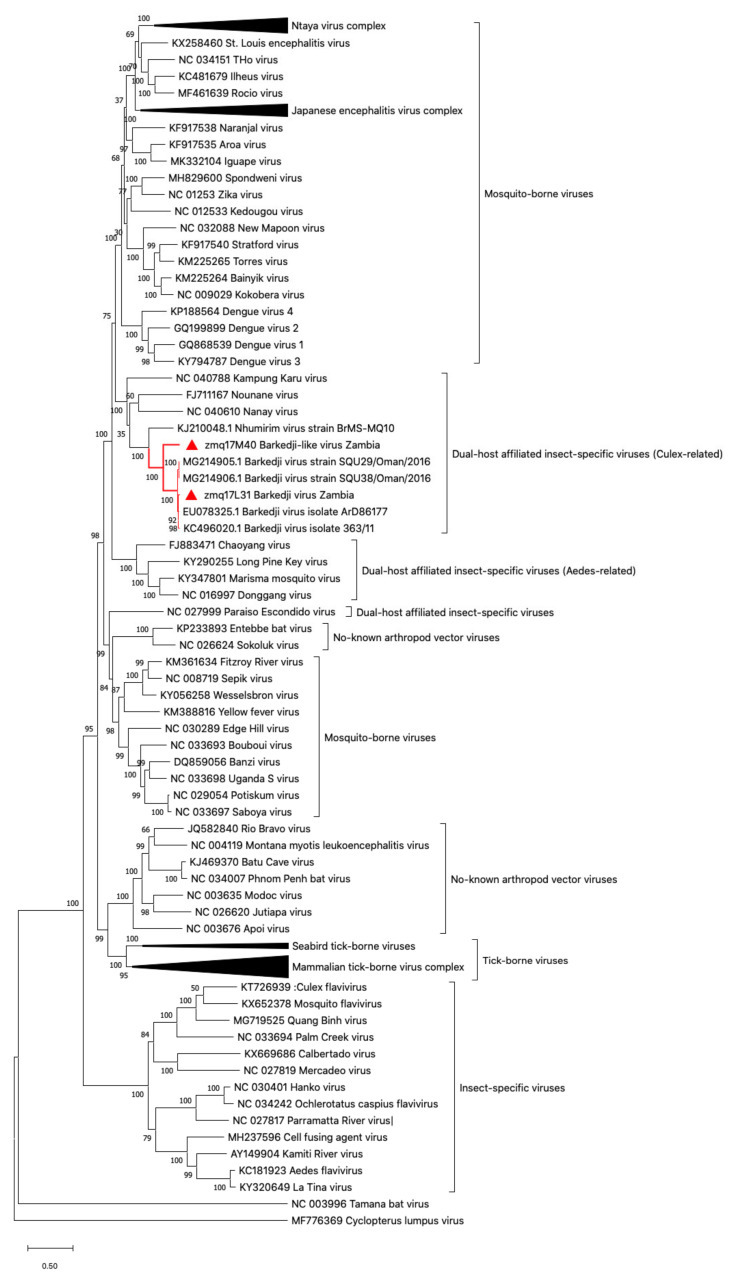
Phylogenetic tree of full-length nucleotide sequence flaviviruses. Barkedji virus strain Zambia (zmq17L31) and Barkedji-like virus (zmq17M40) (marked in the red triangle) are clustered between mosquito-borne flaviviruses. The phylogenetic tree was constructed using Mega X software with a maximum likelihood method based on nucleotide sequence. Bootstrap percentage is shown on the nodes.

**Figure 2 viruses-12-01017-f002:**
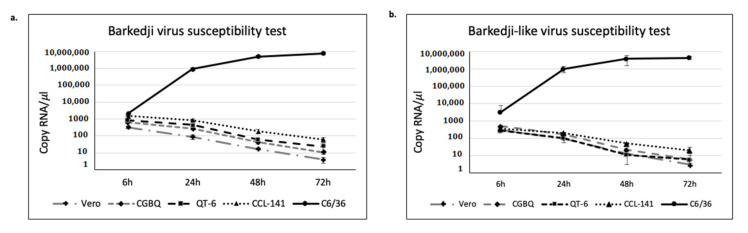
BJV and BJLV growth in several cell lines. BJV (**a**) and BJLV (**b**) infection in mosquito (C6/36), mammalian (Vero) and avian (CCL-141, CGBQ and QT-6) revealed that these viruses only can replicate effectively in mosquito cell line. Data represent the mean (± standard deviation, SD) of three independent experiments.

**Figure 3 viruses-12-01017-f003:**
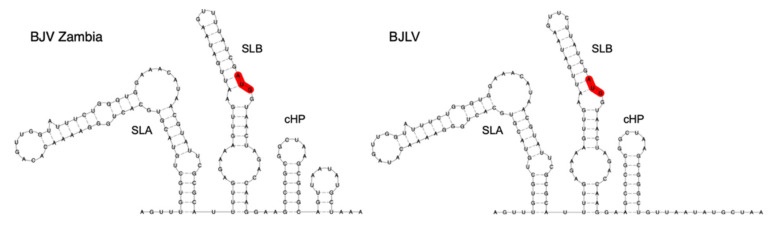
Secondary structure prediction of the genomic upstream regions of BJV Zambia (zmq17L31, **left**) and BJLV (zmq17M40, **right**), comprising 5′-UTRs and the distal parts of the capsid protein nucleotide sequences. The canonical start codon is highlighted in red. Both 5′-UTRs contain the conserved sequence elements SLA and SLB, the latter partially overlapping the beginning of the coding regions. The conserved cis-acting capsid hairpin (cHP) structure is present in both sequences, while a small downstream hairpin is only predicted in BJV Zambia.

**Figure 4 viruses-12-01017-f004:**
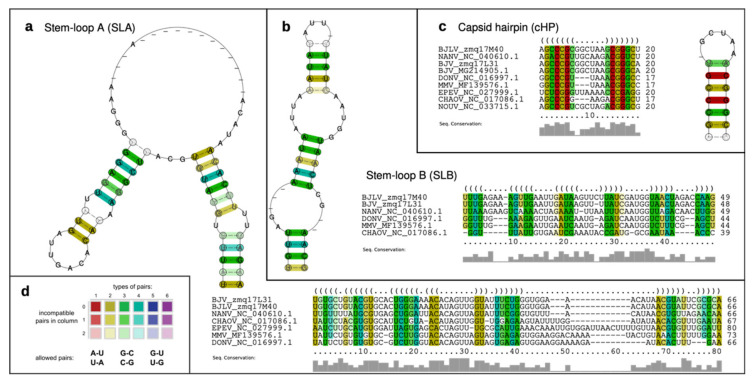
Consensus structures of conserved RNA elements in dISFV 5′-UTRs. (**a**–**c**) Consensus structure prediction of the conserved RNA elements in dISFV 5′-UTRs and adjacent capsid protein coding regions, SLA, SLB and cHP. For each predicted consensus structure, a structural alignment is provided that was built from homologous sequences in related viral species. (**d**) Color coding scheme sensu RNAalifold for paired columns in alignments. Circled nucleotides in the consensus structure plots mark compensatory mutations in the corresponding columns of the alignment, encompassing cases where both positions of a base pair mutated at once, such as GC and CG or GU and UA.

**Figure 5 viruses-12-01017-f005:**
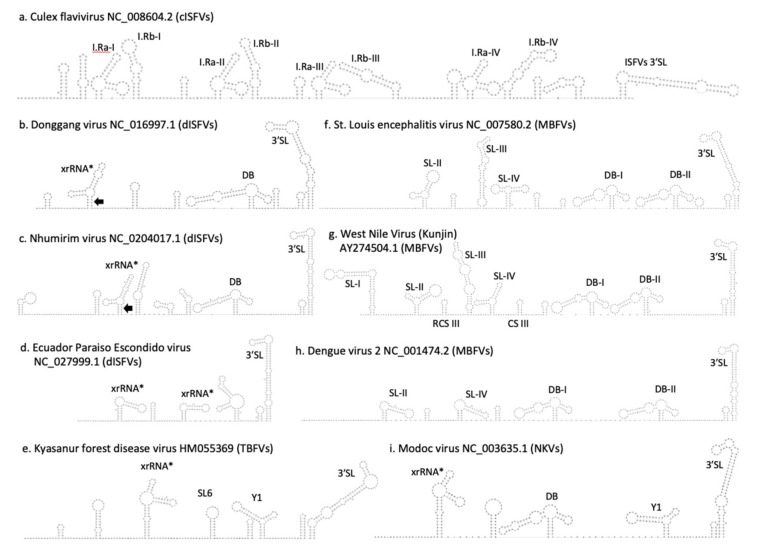
Secondary structure predictions of the 3′-UTRs of different flaviviruses. (**a**) Culex flavivirus NC_008604.2, (**b**) Donggang virus NC_016997.1, (**c**) Nhumirim virus NC_0204017.1, (**d**) Ecuador Paraiso Escondido virus NC_027999.1, (**e**) Kyasanur forest disease virus HM055369, (**f**) St. Louis encephalitis virus NC_007580.2, (**g**) West Nile virus (Kunjin) AY_274504.1, (**h**) Dengue virus 2 NC_001474.2, and (**i**) Modoc virus NC_003635.1. I.Ra/I.Rb: Insect Repeat Element a/b; SL: Stem Loop; DB: Dumbbell; xrRNA: Exoribonuclease resistant RNA; RCS: Repeat Conserved-sequence; CS: Conserved Sequence; T.SL: TBFV SL-shaped element; Y1:TBEV Y1; Black arrow: closing stem. Asterisks mark: in silico predicted xrRNA-like structures, whose exoribonuclease stalling capacity has not been experimentally verified.

**Figure 6 viruses-12-01017-f006:**
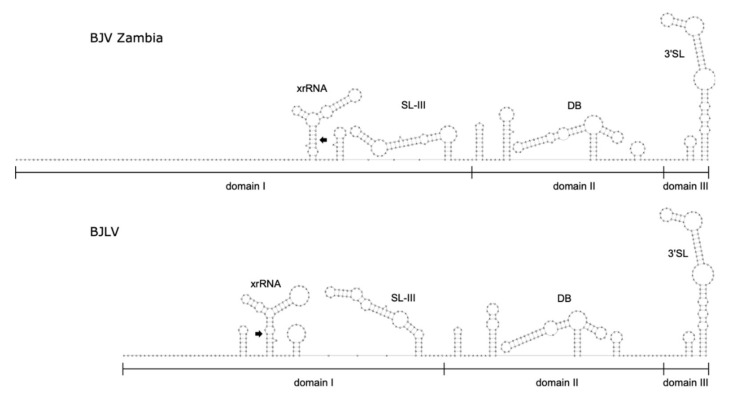
Comparison of BJV/BJLV 3′-UTRs. Secondary structure prediction of the 3′-UTRs of BJV Zambia (zmq17L31, **top**) and BJLV (zmq17M40, **bottom**), encompassing structural homologs to exoribonuclease-resistant RNAs (xrRNA), dumbbell (DB), and long terminal 3′ stem-loop (3′SL) structures, respectively, in domains I-III. A novel conserved stem-loop structure (SL-III), which has not been reported in dISFVs before, is located between the xrRNA and DB elements. Black arrow: closing stem.

**Figure 7 viruses-12-01017-f007:**
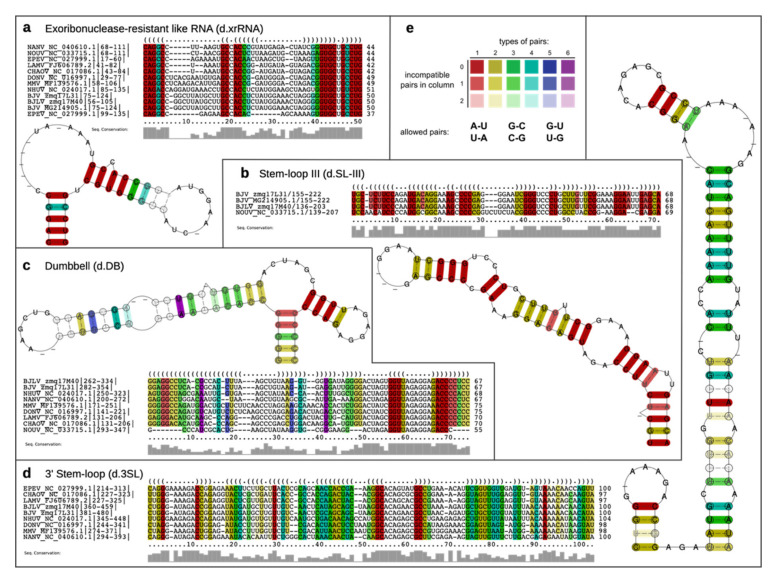
Consensus structure prediction of the conserved RNA elements in dISFV 3′-UTRs. (**a**–**d**) Structural alignments and consensus structure prediction of the conserved elements d.xrRNA, d.SL-III, d.DB and d.3SL (d indicating dISFVs). The xrRNA-like consensus structure shown in (**a**) encompasses the apical three-way junction of BJV and BJLV, as shown in Figure 5. Grey bars below the structural alignments indicate for each column the level of sequence conservation. (**e**) RNAalifold coloring scheme; see Figure 4 for details.

**Figure 8 viruses-12-01017-f008:**
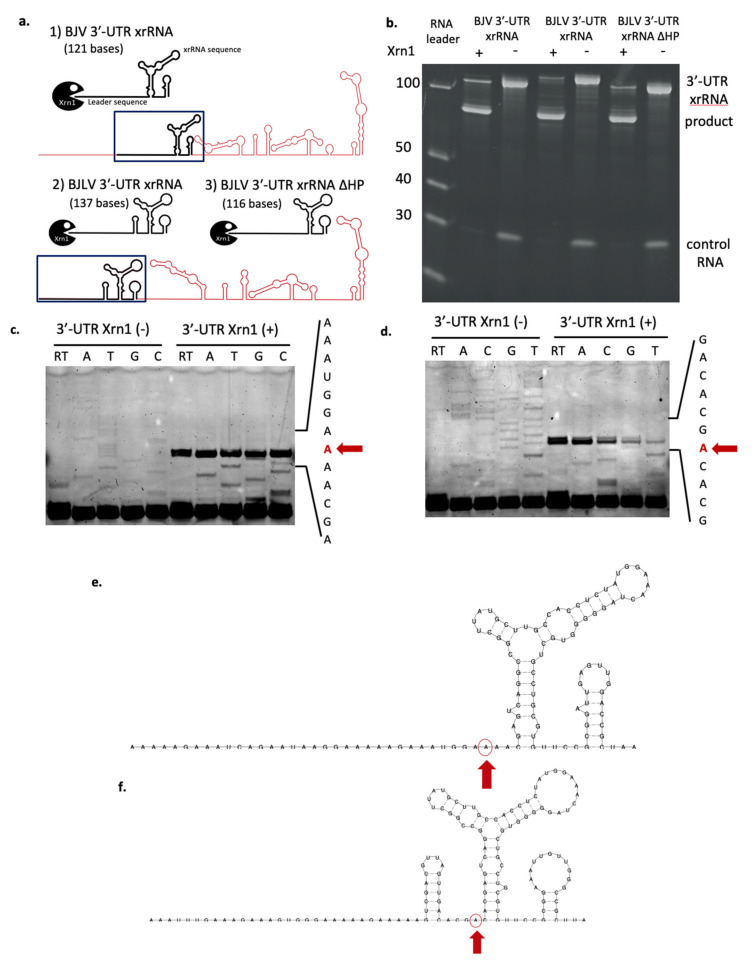
Xrn1 degradation assay and determination of Xrn1 stop point. (**a**) Schematic diagram of the BJV and BJLV 3′-UTR xrRNA degradation assay. xrRNA structures (shown in the box) were challenged with Xrn1 enzyme. (1) BJV 3′-UTR xrRNA (2) BJLV 3′-UTR xrRNA (3) BJV 3′-UTR xrRNA ΔHP. (**b**) In vitro Xrn1 degradation assay against xrRNA structure of BJV and BJLV, and the hairpin structure of BJLV. Data show a representative result of two independent experiments. Examination of BJV (**c**,**e**) and BJLV (**d**,**f**) Xrn1 stop points, which we could unambiguously locate directly at upstream of the xrRNA structures. Red arrow: Stop point.

**Figure 9 viruses-12-01017-f009:**
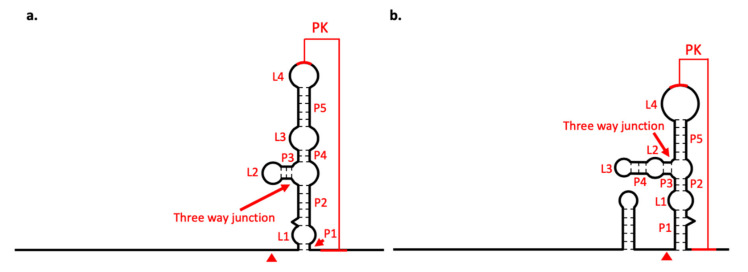
Diagrams of xrRNA structure. xrRNA structure of (**a**) BJV and (**b**) BJLV. Arrows indicate the three-way junction structure and arrowheads indicate Xrn1 stop point; PK: pseudoknot.

**Figure 10 viruses-12-01017-f010:**
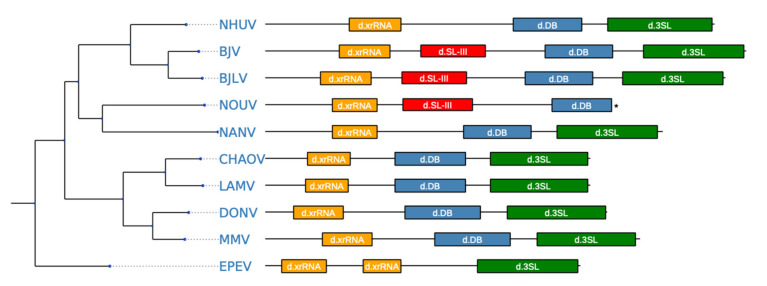
Annotated 3′-UTRs of dISFV species. The tree on the left constitutes a maximum-likelihood phylogenetic tree computed from complete coding sequence nucleotide alignments of all dISFV species with available 3′-UTR sequences. An annotated sketch of the 3′-UTR is plotted to scale for each species. Structural homologs of evolutionarily conserved RNA elements are depicted in colored boxes. Identifiers within the boxes reference elements outlined in Figure 6. The 3′-UTR sequence of Nounane virus (NOUV) is incomplete and marked with an asterisk.

**Table 1 viruses-12-01017-t001:** Number of mosquito species and pools collected in 2017.

Year	Month	Place	Species	Mosquito no.	Pool no.	No. of Flavivirus Positive Pools (%) *	Nucleotide Identity with BJV Israel
2017	April	Livingstone	*Culex quinquefasciatus*	780	34	1 (2.9) ^a^	99%
			*Culex univittatus ^+^*	15	3		
			*Culex nebulosus ^+^*	5	3		
			*Culex tigripes ^+^*	11	3		
			*Culex bitaeniorhynchus ^+^*	1	1		
			*Aedes aegypti*	5	5		
			*Aedes* spp. *^+^*	7	3		
			*Mansonia* sp.	2	2		
			*Anopheles* spp. *^+^*	72	15		
			**Total**	**898**	**69**	1	
2017	May	Mongu	*Culex quinquefasciatus*	285	22		
			*Culex univittatus ^+^*	309	15	2 (13.3) ^b^	81%
			*Culex annulioris ^+^*	41	5	1 (20) ^c^	81%
			*Culex tigripes ^+^*	2	2		
			*Culex spp. ^+^*	109	5		
			*Aedes macintoshi ^+^*	10	3		
			*Aedes aegypti*	1	1		
			*Coquillettidia aurites ^+^*	9	2		
			*Coquillettidia metarika ^+^*	7	2		
			*Coquillettidia fuscopenata ^+^*	46	5		
			*Mansonia* sp.	132	8		
			*Uranotaenia* sp. *^+^*	1	1		
			*Aedeomya* sp.	10	2		
			*Anopheles* spp. *^+^*	1444	65		
			**Total**	**2406**	138	3	

* Positive pools: ^a^ zmq17L31; ^b^ zmq17M54 and zmq17M115; ^c^ zmq17M40. ^+^ Mosquito species was confirmed by sequencing of COI genes in this study.

**Table 2 viruses-12-01017-t002:** Cleavage site prediction of Barkedji (BJV) and Barkedji-like virus (BJLV).

**Virus**	**C/Anch C**	**Anch C/prM**	**PrM/M**	**M/E**
BJV Israel	KTSKR/GLQQS	TMAAC/ATLGM	RRSKR/SVAIA	APAYS/LHCSR
BJV Oman	KTSKR/GLQQS	TMAAC/ATLGM	RRSKR/SVAIA	APAYS/LHCSR
BJV Zambia	KTSKR/GLQQS	TMAAC/ATLGM	RRSKR/SVAIA	APAYS/LHCSR
BJLV	RTAKR/GLGQS	TMAAC/ATLGM	RRSKR/SVAIA	APAYS/LHCAR
**Virus**	**E/NS1**	**NS1/NS2A**	**NS2A/NS2B**	**NS2B/NS3**
BJV Israel	TTVAG/DVGCN	SWTTA/GNATG	GSGKR/SVSMG	KGTQK/AGAMW
BJV Oman	TTVAG/DVGCN	SWTTA/GNATG	GSGKR/SVSMG	KGTQK/AGAMW
BJV Zambia	TTVAG/DVGCN	SWTTA/GNATG	GSGKR/SVSMG	KGTQK/AGAMW
BJLV	TTVAG/DVGCN	SWSTA/GNISG	AAGRR/SVSMG	KPAQK/AGAMW
**Virus**	**NS3/NS4A**	**NS4A/2K**	**2K/NS4B**	**NS4B/NS5**
BJV Israel	AEGRR/GASDI	AEKQR/SAIDN	LAVTA/NEKGL	KSARK/GTPGG
BJV Oman	AEGRR/GASDI	AEKQR/SAIDN	LAVTA/NEKGL	KSARK/GTPGG
BJV Zambia	AEGRR/GASDI	AEKQR/SAIDN	LAVTA/NEKGL	KSARK/GTPGG
BJLV	AEGRR/GAHDL	AEKQR/SAIDN	LAVTA/NEKGL	KSARK/GTPGG

**Table 3 viruses-12-01017-t003:** Pairwise amino acid (aa) comparison based on amino acid sequences between BJLV, BJV Zambia, and Nhumirim virus.

**Virus**	**AnchC-C**		**prM**		**M**		**Envelope**		**NS1**		**NS2A**	
	**aa**	**ID (%)**	**aa**	**ID (%)**	**aa**	**ID (%)**	**aa**	**ID (%)**	**aa**	**ID (%)**	**aa**	**ID (%)**
BJLV	127	-	92	-	75	-	503	-	350	-	233	-
BJV Zambia	125	86.6	992	91.3	75	88.0	503	90.4	350	92.2	233	81.5
Nhumirim virus	128	67.9	94	64.5	75	72.0	503	64.7	351	71.4	232	66.8
**Virus**	**NS2B**		**NS3**		**NS4A**		**NS4B**		**NS5**		**TOTAL**	
	**aa**	**ID (%)**	**aa**	**ID (%)**	**aa**	**ID (%)**	**aa**	**ID (%)**	**aa**	**ID (%)**	**aa**	**ID (%)**
BJLV	129	-	621	-	126	-	255	-	905	-	3439	-
BJV Zambia	129	90.6	621	92.4	126	84.9	255	91.7	905	92.1	3437	90.6
Nhumirim virus	130	63.9	622	74.4	149	62.6	255	73.7	907	77.4	3446	71.2

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
