# Peer review of "Discoveries of Exoribonuclease-Resistant Structures of Insect-Specific Flaviviruses Isolated in Zambia"

_viruses, 2020, doi:10.3390/v12091017_

Round 1

Reviewer 1 Report

Overall this was a well written manuscript that was scientifically sound. I only have a few minor comments:

1. Table 1 is a little hard to read because some of the names in the species column extend to two lines thereby shifting everything else in relation. This minor formatting error should be fixed.

2. A more thorough description of dual-host affiliated viruses should be included in the introduction. What does it mean to be dual host affiliated?

3. Section 3.5 in the results doesn’t show any results. This section should be merged with the M&M description about predicting evolutionarily conserved RNA structures.

Author Response

August 8th, 2020

Manuscript ID viruses-883990

Title: Discoveries of exoribonuclease-resistant structures of insect-specific flaviviruses isolated in Zambia.

Response to Reviewer 1 Comments

Point 1: Table 1 is a little hard to read because some of the names in the species column extend to two lines thereby shifting everything else in relation. This minor formatting error should be fixed. 

Response 1: We appreciate the reviewer’s comment. We have corrected the formatting error and modified Table 1 (lines 285-287, page 7).

Point 2: A more thorough description of dual-host affiliated viruses should be included in the introduction. What does it mean to be dual host affiliated?

Response 2: We apologise for our poor description of dual host affiliated mosquito-borne viruses (dISFVs). dISFVs are phylogenetically closely related to the mosquito-borne flaviviruses (MBFV) and should have the capacity to infect vertebrates, even though it has as yet not been demonstrated that they can either infect vertebrates or cell lines derived from them. We have now provided clearer information on the dISFVs in the Introduction (lines 54-60, page 2) as follows: “Similarly, several dISFVs have been discovered and some of them have been isolated and characterized in mosquitoes. Although dISFVs only replicate well in mosquito-derived cell lines, they are phylogenetically more closely related to MBFVs than cISFVs [4]. Kenney et al. [5] hypothesized about three possibilities for the emergence of dISFVs: First, these viruses represent a distinct group of ISFVs, which did not evolve an ability to replicate in a vertebrate host; second, these viruses were part of a dual-host mosquito-vectored virus but have lost their ability to infect vertebrates; third, these viruses were part of the dual-host mosquito-vectored clade but the non-mosquito secondary host has not yet been identified.”

Point 3: Section 3.5 in the results doesn’t show any results. This section should be merged with the M&M description about predicting evolutionarily conserved RNA structures.

Response 3: Following the reviewer’s suggestion, we have combined subheading 3.5 into subheading 2.9 at the Materials and Methods section as follows:

2.9. Characterization of structurally homologous RNAs

We employed an in silico comparative genomics assay, combining thermodynamic modeling by single sequence folding and consensus structure prediction as well as homology detection by utilizing covariance models (CMs). CMs are statistical models of RNA structure that extend classic Hidden-Markov-Models (HMMs) to represent sequence and secondary structure simultaneously [41]. They provide a powerful mechanism for the identification and characterization of homologous RNA structures and allow for a rapid screening of large RNA sequence databases to find even weakly conserved sequence-only or structurally homologous RNAs [42]. (lines 201-208, page 5).  

Reviewer 2 Report

Following flavivirus screening in field-collected mosqutoes from Zambia, the authors describe isolation and genome sequencing of a local dual-host insect flavivirus isolate and a related novel virus, called as Barkedji-like virus. Moreover, they test cell lines of various origins for susceptibility, characterize extreme non-coding ends of the viral genome and present a detailed comperative evaluation of these regions, supported in part by in vitro resistance assay. The manuscript is rather long but informative, with detailed information and insights into 5’-3’ends of flaviviruses. This reviewer has few minor recommendations and encourage publication following revision.

(page 1 line 41): Bunyavirales is an order, please rephrase.

(page 2 lines 61-62): The statement implies Aedes mosquitoes as vectors for hemorrhagic fever-associated viruses but they also transmit the well-known Zika virus. Please rephrase.   

(page 2 lines 72 and elsewhere): evolutionarily-conserved?

(page 3 lines 103-104): “Viral genome sequences and their susceptibility...” Please rephrase.  

(Results): Please provide statements on findings of the COI PCR for mosquito identification.

(Table 1): Please align rows so that the identity percentages are displayed in line with positive pool data. 

(page 10 line 318): Delete extra space before “Production of...”

(page 10 lines 325-332): The subheading 3.5 merely describes the approach and is basically explanatory, without presenting any findings. I am not sure whether it is justified as a stand-alone section in results.

Author Response

August 8th, 2020

Manuscript ID viruses-883990

Title: Discoveries of exoribonuclease-resistant structures of insect-specific flaviviruses isolated in Zambia.

Response to Reviewer 2 Comments

Point 1: (page 1 line 41): Bunyavirales is an order, please rephrase. 

Response 1: We appreciate the reviewer’s comment. We have modified the sentence as follows: “These belong to a range of viruses in the order Bunyavirales and several families, including Birnaviridae, Togaviridae, Rhabdoviridae, Reoviridae, and Flaviviridae.” (lines 40-41, page 1).

Point 2: page 2 lines 61-62): The statement implies Aedes mosquitoes as vectors for hemorrhagic fever-associated viruses but they also transmit the well-known Zika virus. Please rephrase.

Response 2: Following the reviewer’s suggestion, we have modified the sentence as follows: “The Aedes spp. mosquitoes are vectors for clinically important flaviviruses, such as DENV, Yellow fever virus (YFV) and ZIKV.” (lines 65-66, page 2).

Point 3: (page 2 lines 72 and elsewhere): evolutionarily-conserved?

Response 3: Following the reviewer’s suggestion, we have modified this phrase (lines 76 and 85, page 2; line 105, Page 3; line 366, page 11; line 531, page 17).

Point 4: (page 3 lines 103-104): “Viral genome sequences and their susceptibility...” Please rephrase.  

Response 4: According to the reviewer’s suggestion, we have modified the entire paragraph as follows: “In this study, we report that our surveillance studies discovered two dISFVs, Barkedji virus (BJV) and a new virus tentatively named Barkedji-like virus (BJLV) from Culex mosquitoes. In parallel, to understand the evolutionary traits of dISFVs, we characterized the evolutionarily-conserved, structural RNA elements in the untranslated regions of BJV and BJLV in silico by single sequence folding and consensus structure prediction. Furthermore, we focused on the functional characterization of BJV and BJLV xrRNA structures.” (lines 103-108, page 3).

Point 5: (Results): Please provide statements on findings of the COI PCR for mosquito identification.

Response 5: Following the reviewer’s suggestion, we have added mark (+) at the species that we determined through genomic analysis in Table 1 (lines 285-287, page 7).

Point 6: (Table 1): Please align rows so that the identity percentages are displayed in line with positive pool data. 

Response 6: We appreciate the reviewer’s comment. We have corrected the formatting error and modified Table 1 (lines 285-287, page 7)

Point 7: (page 10 line 318): Delete extra space before “Production of...”

Response 7: We have modified the sentence as the reviewer suggested (line 322, page 10).

Point 8: (page 10 lines 325-332): The subheading 3.5 merely describes the approach and is basically explanatory, without presenting any findings. I am not sure whether it is justified as a stand-alone section in results.

Response 8: Following the reviewer’s suggestion, we have combined subheading 3.5 into subheading 2.9 at the Materials and Methods section as follows:

2.9. Characterization of structurally homologous RNAs

We employed an in silico comparative genomics assay, combining thermodynamic modeling by single sequence folding and consensus structure prediction as well as homology detection by utilizing covariance models (CMs). CMs are statistical models of RNA structure that extend classic Hidden-Markov-Models (HMMs) to represent sequence and secondary structure simultaneously [41]. They provide a powerful mechanism for the identification and characterization of homologous RNA structures and allow for a rapid screening of large RNA sequence databases to find even weakly conserved sequence-only or structurally homologous RNAs [42]. (lines 201-208, page 5)

Reviewer 3 Report

Review viruses 883990

In the manuscript: “Discoveries of exoribonuclease-resistant structures of insect-specific flaviviruses isolated in Zambia”, the authors describe the discovery of a novel lineage 2 or dual host affiliated insect-specific flavivirus (dISFV) that is related to Barkedji virus (BJV), termed Barkedji-like virus (BJLV).  This virus, isolated from mosquitoes, was inoculated on mosquito and vertebrate cell types from a small selection of animal origins and only replicated on C6/36 mosquito cells. In silico analyses revealed that both BJV and BJLV contained RNA structures in the 5’ and the 3’ UTRs. The authors then continue with in vitro experiments to show that some of the RNA structures in the 3’ UTR can halt the activity of exonuclease XRN1.

Although xrRNA structures in the flavivirus 3’ UTR is a fascinating field of research and the analyses and experiments in this manuscript seem interesting, there are a number of major concerns that need to be adequately addressed before this study is fit for publication. Most importantly, it needs to be clear why experiments are performed and how they should be interpreted.  

Major points:

  1. Viruses in the genus Flavivirus have a highly structured 3’UTR which contains structures that can stall host exonuclease XRN1. After reading the complete manuscript, it is still unclear what the novelty of this study is and why after isolating a novel virus investigating xrRNA structures was of interest to the authors.
  2. From the abstract and introduction it remains unclear what the exact aim of the study is. In the results section the choice for specific experiments is also not explained. For instance; the relationship between identification of the SLIII in BJV and BJLV and the xrn1 assay in figure 8 is unclear as figure 8 does not test the functionality of SLIII.
  3. Although most of the text is easy to read, it is incredibly difficult to interpret. This is most clear in the discussion, which, in my opinion, is not publishable in its current form. Most of the discussion is filled with seemingly strange arguments that should either be clarified or deleted (see specific comments below). The main example is a very wild hypothesis that dISFs have unidentified vertebrate hosts, while their own experimental data and many previous studies strongly indicated that ISFs are indeed insect-specific. This single unfounded discussion point goes on for a full page, while the authors have only put minimal effort into discovering this hypothesized vertebrate host organism (figure 1), which makes their fixation on this point the more surprising.
  4. Instead the authors should focus on discussion points that are more relevant to the presented results. Questions such as; what features of SLIII or the xrRNA structure in BJV and BJLV make them stand out from other dISFs, what is the biological role of the xrRNA structure, the presence of sfRNA, inhibition of antiviral responses and potential for transmission dynamics are currently not discussed and could strengthen the discussion.

Specific points:

Line 54: “ Similarly, several dISFVs have been discovered and some of them have been isolated and characterized in mosquitoes.” Perhaps this sentence is just a bit awkwardly written, however it suggests to me that other dISFs have been isolated from sources different from mosquitoes or can be characterised (i.e. replicate) in cells from vertebrate hosts. The authors do not elaborate on this, nor do I think there is much experimental evidence to support this. Please either revise or elaborate.

Line 83: “appear with different copy numbers”, rephrase. ‘Appear’ is an active verb while genetic elemenets are either present or they are not, and ‘copy numbers’ generally refers to genome copies which may be confusing when talking about specific duplicated genetic elements

Line 83-85: “Importantly, some secondary structure elements in flavivirus 3-UTRs can block RNA degradation by endogenous 5ï‚¢-3ï‚¢ exoribonucleases, such as XRN-1, resulting in the accumulation of sub-genomic flavivirus RNAs (sfRNAs) in the infected cells. The latter have been shown to interfere with the host immune responses, thereby modulating viral pathogenesis [22].” Unclear what ‘some’ and ‘the latter’ refer to.

In the following paragraph (Line 88-94) I think the goal is to make clear that regions of Zambia have sufficient mosquito populations to support outbreaks of arboviral disease. However the paragraph vaguely describes distinctions in ecosystems without providing information or evidence that competent vectors are present in large numbers. Also, the next paragraph (specifically Lines 96-97) provides much better evidence for the risk of flavivirus infections in Zambia. I would therefore suggest to delete Lines 88-94 from the introduction as it has little added value.  

Paragraph 3.5 introduces the method, but does not display any results. This section of text should therefore not have a separate heading, as it merely functions to introduce the analysis done in following paragraphs.

Line  334: “the upstream regions of both BJV Zambia and BJLV” unclear what the upstream region of a virus is...

Lines 401-403: “Interestingly, among dISFVs the SL-III element is only conserved in BJV, BJLV, and NOUV. A consensus structure prediction of SL-III within dISFVs is shown in Figure 7b” The alignment in figure 7b only displays sequences from BJV, BJLV and NOUV. This statement would be strengthened if the authors additionally displayed sequences from other dISFs to show they do not contain a sequence fitting for the same structural element.  

Lines 467-484 including the following sentences: “These findings have led us to speculate on possible routes of the BJV dissemination. We consider that there are at least three possibilities: First, BJV infects birds that can migrate easily within Africa and Asia [64]. Second, the virus may have been originally endemic in these different regions but was not been identified before 2007. Lastly, mosquitoes infected with BJV could have been unintentionally transported to different countries through international air travels or overseas routes [65].” 

These speculations come completely out of the blue and seem mostly unfounded. I do not know why dissemination (presumably meant as global distribution / spread) needs to be discussed here. Perhaps more importantly, BJV is an insect-specific flavivirus (ISFV) and the experiments described in this study confirm the insect-specific nature of BJV. This entire paragraph is therefore dedicated to a wild and highly unlikely scenario where this ISFV infects birds, while there were no reasonable clues to suggest that this could be the case nor is there any evidence presented by the authors that warrants this discussion. As the result that insect-specific viruses only replicate in insect cells only states the obvious, these first paragraphs (lines 267 to 484) should be removed.   

The following paragraphs (lines 484-502) continue discussing the possibility that BJV or BJLV are arboviruses. Though general hypotheses on dISF evolution are more reasonably explained here, the discussion on phylogeny is chaotic and the authors keep hinting that there should be a vertebrate host somewhere (“but not as yet known to be present in vertebrate hosts” and “As both BJV/BJLV and WNV isolated in Mongu have been identified in Culex mosquitoes with low infection rates [31], unlike cISFVs such as Culex flavivirus with high infection rates among Culex quinquefasciatus mosquito populations, these dISFVs seem likely to have other non-mosquito hosts. Further investigations of dISFV-infected species are needed to clarify this.”). Most of this should be deleted as it can be summarised in two sentences, is mostly unfounded and is not even a key subject of the presented experiments.  

Lines 503-516: As the stem loop A and B are known to be strongly conserved among most flaviviruses, the point of this section is unclear. What is novel about BJV and BJLV having an SLA in the 5’ UTR and/or what are the advantages of the used method of analysis over other methods?

Lines 517-518: “BJV and BJLV phylogenetically cluster with NHUV, NOUV, NANV, and KPKV, thereby forming a sister clade to all MBFVs except the Yellow Fever virus complex”. BJV and BJLV are dISFs which is still a clearly separate phylogenetic group. Calling it a sister clade to MBFVs is thus incorrect.

Line 524: Unclear. Which viruses are being compared? What does “these viruses” refer to?

Lines 526-528: “Experimental verification of the predicted exoribonuclease-stalling activity revealed that the xrRNA elements of BJV and BJLV could halt degradation by Xrn 1”. The study would benefit from some explanation by the authors why this observation is novel or interesting.

Lines 529-536: In this paragraph the authors use a previously described classification for xrRNA structures (class 1 and 2) and discuss why the xrRNA in BJV and BJLV is a class 1 xrRNA. However, it is unclear from this manuscript what the difference between class 1 and class 2 xrRNA structures is. Also dISFs supposidly have class 2 xrRNA structures, but the authors do not discuss why dISFs BJV and BJLV are different in that they contain a class 1 xrRNA.

Line 552: “similarly”. Similar to what? Seems to be unconnected.  

Round 2

Reviewer 3 Report

The manuscript has been edited extensively by the authors and in my opinion it now describes the aim and novelty of the study more clearly. Although the findings are perhaps not the most surprising, confirming the functionality of xrRNA in dISFs is an important step towards better understanding the role of flavivirus sfRNA and flavivirus evolution. The subjects raised in the new version of the discussion are also more relevant and fit better with the molecular character of the study. 

The study does rely heavily on modeling of RNA structures, which makes it the more important that the few wet-lab experiments (figures 2 and 8) are adequately described. Figure 2 - virus growth curves- can the authors describe what the data points are based on (i.e. How many experimental repeats and duplicate samples?). Preferably then also supply error bars to the figure. Graphically figure 2 is also not very appealing. 

Figure 8 is a qualitative figure, and is fine the way it is presented. However, please do list in the text of figure legend whether this result has been confirmed with experimental repeats.

Final suggestion: Line: 585-586: "Further investigation is needed to investigate the formation of the sfRNA and their molecular interactions with host factors"  There are a few studies that do report molecular interactions between flavivirus sfRNA and mosquito host proteins that may also influence dISF replication in the mosquito host. Perhaps the following paper can be of use: https://doi.org/10.1073/pnas.190561711

Author Response

August 13, 2020

Manuscript ID viruses-883990

Title: Discoveries of exoribonuclease-resistant structures of insect-specific flaviviruses isolated in Zambia.

Response to the Comments of Reviewer 3

The study does rely heavily on modeling of RNA structures, which makes it the more important that the few wet-lab experiments (figures 2 and 8) are adequately described.

Point 1: Figure 2 - virus growth curves- can the authors describe what the data points are based on (i.e. How many experimental repeats and duplicate samples?). Preferably then also supply error bars to the figure. Graphically figure 2 is also not very appealing. 

Response 1: We appreciate the reviewer’s comment and apologize for the lack of error bars in the Figure 2. We have repeated the virus growth curve experiments twice. The graphs in Figure 2 were based on the representative data of similar results. We have added the sentence in the Figure legend of Figure 2 as follows: “Graphs show representative data of two independent experiments.” (lines 328-329, page 10).

We are now carrying out as requested by the Reviewer the virus growth curve experiments again to include error bars to each point of data in Figure 2.

Point 2: Figure 8 is a qualitative figure, and is fine the way it is presented. However, please do list in the text of figure legend whether this result has been confirmed with experimental repeats.

Response 2: Following the reviewer’s suggestion, we have modified the sentence in the Figure legend of Figure 8 as follows: “Data shows a representative result of two independent experiments “ (line 458, page 15).

Point 3: Final suggestion: Line: 585-586: "Further investigation is needed to investigate the formation of the sfRNA and their molecular interactions with host factors"  There are a few studies that do report molecular interactions between flavivirus sfRNA and mosquito host proteins that may also influence dISF replication in the mosquito host. Perhaps the following paper can be of use: https://doi.org/10.1073/pnas.190561711

Response 3: Following the reviewer’s suggestion, we have modified the sentence and add reference [65] (Göertz, G.P. et al. Subgenomic flavivirus RNA binds the mosquito DEAD/H-box helicase ME31B and determines Zika virus transmission by Aedes aegypti. Proc Natl Acad Sci U S A 2019, 116, 19136-19144.) as follows: “The presence of functional xrRNA in BJV and BJLV suggest that dISFVs have potency to produce sfRNA which could alter the cellular immune response and promote flavivirus transmission by mosquitoes [65]. Further investigation is needed to investigate the formation of the sfRNA and their molecular interactions with host factors in dISFVs.” (lines 505-509, page 16).